# DEEP AUTOAUGMENT

**Yu Zheng[1], Zhi Zhang[2], Shen Yan[1], Mi Zhang[1]**
[1]Michigan State University,    [2]Amazon Web Services
zhengy30@msu.edu, zhiz@amazon.com, {yanshen6, mizhang}@msu.edu

## ABSTRACT

While recent automated data augmentation methods lead to state-of-the-art results, their design spaces and the derived data augmentation strategies still incorporate strong human priors. In this work, instead of fixing a set of hand-picked default augmentations alongside the searched data augmentations, we propose a fully automated approach for data augmentation search named **Deep AutoAugment** (**DeepAA**). DeepAA progressively builds a multi-layer data augmentation pipeline from scratch by stacking augmentation layers one at a time until reaching convergence. For each augmentation layer, the policy is optimized to maximize the cosine similarity between the gradients of the original and augmented data along the direction with low variance. Our experiments show that even without default augmentations, we can learn an augmentation policy that achieves strong performance with that of previous works. Extensive ablation studies show that the regularized gradient matching is an effective search method for data augmentation policies. Our code is available at: https://github.com/MSU-MLSys-Lab/DeepAA.

## 1 INTRODUCTION

Data augmentation (DA) is a powerful technique for machine learning since it effectively regularizes the model by increasing the number and the diversity of data points (Goodfellow et al., 2016; Zhang et al., 2017). A large body of data augmentation transformations has been proposed (Inoue, 2018; Zhang et al., 2018; DeVries & Taylor, 2017; Yun et al., 2019; Hendrycks et al., 2020; Yan et al., 2020) to improve model performance. While applying a set of well-designed augmentation transformations could help yield considerable performance enhancement especially in image recognition tasks, manually selecting high-quality augmentation transformations and determining how they should be combined still require strong domain expertise and prior knowledge of the dataset of interest. With the recent trend of automated machine learning (AutoML), data augmentation search flourishes in the image domain (Cubuk et al., 2019; 2020; Ho et al., 2019; Lim et al., 2019; Hataya et al., 2020; Li et al., 2020; Liu et al., 2021), which yields significant performance improvement over hand-crafted data augmentation methods.

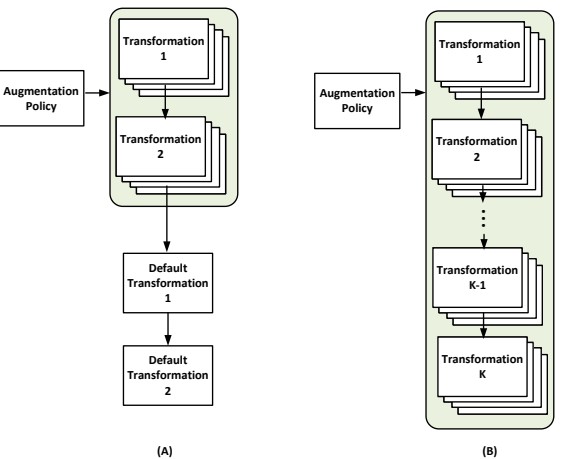

Figure 1: (A) Existing automated data augmentation methods with shallow augmentation policy followed by hand-picked transformations. (B) DeepAA with deep augmentation policy with no hand-picked transformations.

Although data augmentation policies in previous works (Cubuk et al., 2019; 2020; Ho et al., 2019; Lim et al., 2019; Hataya et al., 2020; Li et al., 2020) contain multiple transformations applied sequentially, only one or two transformations of each sub-policy are found through searching whereas the rest transformations are hand-picked and applied by default in addition to the found policy (Figure 1(A)). From this perspective, we believe that previous automated methods are *not entirely automated* as they are still built upon hand-crafted default augmentations.

In this work, we propose Deep AutoAugment (DeepAA), a multi-layer data augmentation search method which aims to remove the need of hand-crafted default transformations (Figure 1(B)). DeepAA fully automates the data augmentation process by searching a deep data augmentation policy on an expanded set of transformations that includes the widely adopted search space and the default transformations (*e.g.* flips, Cutout, crop). We formulate the search of data augmentation policy as a regularized gradient matching problem by maximizing the cosine similarity of the gradients between augmented data and original data with regularization. To avoid exponential growth of dimensionality of the search space when more augmentation layers are used, we incrementally stack augmentation layers based on the data distribution transformed by all the previous augmentation layers.

We evaluate the performance of DeepAA on three datasets – CIFAR-10, CIFAR-100, and ImageNet – and compare it with existing automated data augmentation search methods including AutoAugment (AA) (Cubuk et al., 2019), PBA (Ho et al., 2019), Fast AutoAugment (FastAA) (Lim et al., 2019), Faster AutoAugment (Faster AA) (Hataya et al., 2020), DADA (Li et al., 2020), RandAugment (RA) (Cubuk et al., 2020), UniformAugment (UA) (LingChen et al., 2020), TrivialAugment (TA) (Müller & Hutter, 2021), and Adversarial AutoAugment (AdvAA) (Zhang et al., 2019). Our results show that, without any default augmentations, DeepAA achieves the best performance compared to existing automatic augmentation search methods on CIFAR-10, CIFAR-100 on Wide-ResNet-28-10 and ImageNet on ResNet-50 and ResNet-200 with standard augmentation space and training procedure.

We summarize our main contributions below:

- We propose Deep AutoAugment (DeepAA), a fully automated data augmentation search method that finds a multi-layer data augmentation policy from scratch.

- We formulate such multi-layer data augmentation search as a regularized gradient matching problem. We show that maximizing cosine similarity along the direction of low variance is effective for data augmentation search when augmentation layers go deep.

- We address the issue of exponential growth of the dimensionality of the search space when more augmentation layers are added by incrementally adding augmentation layers based on the data distribution transformed by all the previous augmentation layers.

- Our experiment results show that, without using any default augmentations, DeepAA achieves stronger performance compared with prior works.

## 2 RELATED WORK

**Automated Data Augmentation.** Automating data augmentation policy design has recently emerged as a promising paradigm for data augmentation. The pioneer work on automated data augmentation was proposed in AutoAugment (Cubuk et al., 2019), where the search is performed under reinforcement learning framework. AutoAugment requires to train the neural network repeatedly, which takes thousands of GPU hours to converge. Subsequent works (Lim et al., 2019; Li et al., 2020; Liu et al., 2021) aim at reducing the computation cost. Fast AutoAugment (Lim et al., 2019) treats data augmentation as inference time density matching which can be implemented efficiently with Bayesian optimization. Differentiable Automatic Data Augmentation (DADA) (Li et al., 2020) further reduces the computation cost through a reparameterized Gumbel-softmax distribution (Jang et al., 2017). RandAugment (Cubuk et al., 2020) introduces a simplified search space containing two interpretable hyperparameters, which can be optimized simply by grid search. Adversarial AutoAugment (AdvAA) (Zhang et al., 2019) searches for the augmentation policy in an adversarial and online manner. It also incorporates the concept of Batch Augmentaiton (Berman et al., 2019; Hoffer et al., 2020), where multiple adversarial policies run in parallel. Although many automated data augmentation methods have been proposed, the use of default augmentations still imposes strong domain knowledge.

**Gradient Matching**. Our work is also related to gradient matching. In (Du et al., 2018), the authors showed that the cosine similarity between the gradients of different tasks provides a signal to detect when an auxiliary loss is helpful to the main loss. In (Wang et al., 2020), the authors proposed to use cosine similarity as the training signal to optimize the data usage via weighting data points. A similar approach was proposed in (Müller et al., 2021), which uses the gradient inner product as a per-example reward for optimizing data distribution and data augmentation under the reinforcement learning framework. Our approach also utilizes the cosine similarity to guide the data augmentation

search. However, our implementation of cosine similarity is different from the above from two aspects: we propose a Jacobian-vector product form to backpropagate through the cosine similarity, which is computational and memory efficient and does not require computing higher order derivative; we also propose a sampling scheme that effectively allows the cosine similarity to increase with added augmentation stages.

# 3 DEEP AUTOAUGMENT

## 3.1 OVERVIEW

Data augmentation can be viewed as a process of filling missing data points in the dataset with the same data distribution (Hataya et al., 2020). By augmenting a single data point multiple times, we expect the resulting data distribution to be close to the full dataset under a certain type of transformation. For example, by augmenting a single image with proper color jittering, we obtain a batch of augmented images which has similar distribution of lighting conditions as the full dataset. As the distribution of augmented data gets closer to the full dataset, the gradient of the augmented data should be steered towards a batch of original data sampled from the dataset. In DeepAA, we formulate the search of the data augmentation policy as a regularized gradient matching problem, which manages to steer the gradient to a batch of original data by augmenting a single image multiple times. Specifically, we construct the augmented training batch by augmenting a single training data point multiple times following the augmentation policy. We construct a validation batch by sampling a batch of original data from the validation set. We expect that by augmentation, the gradient of augmented training batch can be steered towards the gradient of the validation batch. To do so, we search for data augmentation that maximizes the cosine similarity between the gradients of the validation data and the augmented training data. The intuition is that an effective data augmentation should preserve data distribution (Chen et al., 2020) where the distribution of the augmented images should align with the distribution of the validation set such that the training gradient direction is close to the validation gradient direction.

Another challenge for augmentation policy search is that the search space can be prohibitively large with deep augmentation layers ($K \geq 5$). This was not a problem in previous works, where the augmentation policies is shallow ($K \leq 2$). For example, in AutoAugment Cubuk et al. (2019), each sub-policy contains $K = 2$ transformations to be applied sequentially, and the search space of AutoAugment contains 16 image operations and 10 discrete magnitude levels. The resulting number of combinations of transformations in AutoAugment is roughly $(16 \times 10)^2 = 25,600$, which is handled well in previous works. However, when discarding the default augmentation pipeline and searching for data augmentations from scratch, it requires deeper augmentation layers in order to perform well. For a data augmentation with $K = 5$ sequentially applied transformations, the number of sub-policies is $(16 \times 10)^5 \approx 10^{11}$, which is prohibitively large for the following two reasons. First, it becomes less likely to encounter a good policy by exploration as good policies become more sparse on high dimensional search space. Second, the dimension of parameters in the policy also grows with $K$, making it more computational challenging to optimize. To tackle this challenge, we propose to build up the full data augmentation by progressively stacking augmentation layers, where each augmentation layer is optimized on top of the data distribution transformed by all previous layers. This avoids sampling sub-policies from such a large search space, and the number of parameters of the policy is reduced from $|\mathbb{T}|^K$ to $\mathbb{T}$ for each augmentation layer.

## 3.2 SEARCH SPACE

Let $\mathbb{O}$ denote the set of augmentation operations (*e.g.* identity, rotate, brightness), $m$ denote an operation magnitude in the set $\mathbb{M}$, and $x$ denote an image sampled from the space $\mathcal{X}$. We define the set of transformations as the set of operations with a fixed magnitude as $\mathbb{T} := \{t | t = o(\cdot \; ; \; m), \; o \in \mathbb{O} \text{ and } m \in \mathbb{M}\}$. Under this definition, every $t$ is a map $t : \mathcal{X} \to \mathcal{X}$, and there are $|\mathbb{T}| = |\mathbb{M}| \cdot |\mathbb{O}|$ possible transformations. In previous works (Cubuk et al., 2019; Lim et al., 2019; Li et al., 2020; Hataya et al., 2020), a data augmentation policy $\mathcal{P}$ consists of several sub-policies. As explained above, the size of candidate sub-policies grows exponentially with depth $K$. Therefore, we propose a practical method that builds up the full data augmentation by progressively stacking augmentation layers. The final data augmentation policy hence consists of $K$ layers of sequentially applied policy $\mathcal{P} = \{\mathcal{P}_1, \cdots, \mathcal{P}_K\}$, where policy $\mathcal{P}_k$ is optimized conditioned on the data distribution augmented

by all previous $(k-1)$ layers of policies. Thus we write the policy as a conditional distribution $\mathcal{P}_k := p_{\theta_k}(n|\{\mathcal{P}_1, \cdots, \mathcal{P}_{k-1}\})$ where $n$ denotes the indices of transformations in $\mathbb{T}$. For the purpose of clarity, we use a simplified notation as $p_{\theta_k}$ to replace $p_{\theta_k}(n|\{\mathcal{P}_1, \cdots, \mathcal{P}_{k-1}\})$.

### 3.3 AUGMENTATION POLICY SEARCH VIA REGULARIZED GRADIENT MATCHING

Assume that a single data point $x$ is augmented multiple times following the policy $p_\theta$. The resulting average gradient of such augmentation is denoted as $g(x, \theta)$, which is a function of data $x$ and policy parameters $\theta$. Let $v$ denote the gradients of a batch of the original data. We optimize the policy by maximizing the cosine similarity between the gradients of the augmented data and a batch of the original data as follows:

$$\theta = \arg\max_\theta \text{ cosineSimilarity}(v, g(x, \theta)) \tag{1}$$

$$= \arg\max_\theta \frac{v^T \cdot g(x, \theta)}{\|v\| \cdot \|g(x, \theta)\|}$$

where $\|\cdot\|$ denotes the L2-norm. The parameters of the policy can be updated via gradient ascent:

$$\theta \leftarrow \theta + \eta \nabla_\theta \text{ cosineSimilarity}(v, g(x, \theta)), \tag{2}$$

where $\eta$ is the learning rate.

#### 3.3.1 POLICY SEARCH FOR ONE LAYER

We start with the case where the data augmentation policy only contains a single augmentation layer, *i.e.*, $\mathcal{P} = \{p_\theta\}$. Let $L(x; w)$ denote the classification loss of data point $x$ where $w \in \mathbb{R}^D$ represents the flattened weights of the neural network. Consider applying augmentation on a single data point $x$ following the distribution $p_\theta$. The resulting averaged gradient can be calculated analytically by averaging all the possible transformations in $\mathbb{T}$ with the corresponding probability $p(\theta)$:

$$g(x; \theta) = \sum_{n=1}^{|\mathbb{T}|} p_\theta(n) \nabla_w L(t_n(x); w) \tag{3}$$

$$= G(x) \cdot p_\theta$$

where $G(x) = \left[ \nabla_w L(t_1(x); w), \cdots, \nabla_w L(t_{|\mathbb{T}|}(x); w) \right]$ is a $D \times |\mathbb{T}|$ Jacobian matrix, and $p_\theta = [p_\theta(1), \cdots, p_\theta(|\mathbb{T}|)]^T$ is a $|\mathbb{T}|$ dimensional categorical distribution. The gradient w.r.t. the cosine similarity in Eq. (2) can be derived as:

$$\nabla_\theta \text{ cosineSimilarity}(v, g(x; \theta)) = \nabla_\theta p_\theta \cdot r \tag{4}$$

where

$$r = G(x)^T \left( \frac{v}{\|g(\theta)\|} - \frac{v^T g(\theta)}{\|g(\theta)\|^2} \cdot \frac{g(\theta)}{\|g(\theta)\|} \right) \tag{5}$$

which can be interpreted as a reward for each transformation. Therefore, $p_\theta \cdot r$ in Eq.(4) represents the average reward under policy $p_\theta$.

#### 3.3.2 POLICY SEARCH FOR MULTIPLE LAYERS

The above derivation is based on the assumption that $g(\theta)$ can be computed analytically by Eq.(3). However, when $K \geq 2$, it becomes impractical to compute the average gradient of the augmented data given that the search space dimensionality grows exponentially with $K$. Consequently, we need to average the gradient of all $|\mathbb{T}|^K$ possible sub-policies.

To reduce the parameters of the policy to $\mathbb{T}$ for each augmentation layer, we propose to incrementally stack augmentations based on the data distribution transformed by all the previous augmentation layers. Specifically, let $\mathcal{P} = \{\mathcal{P}_1, \cdots, \mathcal{P}_K\}$ denote the $K$-layer policy. The policy $\mathcal{P}_k$ modifies the data distribution on top of the data distribution augmented by the previous $(k-1)$ layers. Therefore, the policy at the $k^{th}$ layer is a distribution $\mathcal{P}_k = p_{\theta_k}(n)$ conditioned on the policies $\{\mathcal{P}_1, \cdots, \mathcal{P}_{k-1}\}$

where each one is a $|\mathbb{T}|$-dimensional categorical distribution. Given that, the Jacobian matrix at the $k^{th}$ layer can be derived by averaging over the previous $(k-1)$ layers of policies as follows:

$$G(x)^k = \sum_{n_{k-1}=1}^{|\mathbb{T}|} \cdots \sum_{n_1=1}^{|\mathbb{T}|} p_{\theta_{k-1}}(n_{k-1}) \cdots p_{\theta_1}(n_1) [\nabla_w L((t_1 \circ t_{n_{k-1}} \cdots \circ t_{n_1})(x); w), \cdots , \tag{6}$$

$$\nabla_w L((t_{|\mathbb{T}|} \circ t_{n_{k-1}} \circ \cdots \circ t_{n_1})(x); w)]$$

where $G^k$ can be estimated via the Monte Carlo method as:

$$\tilde{G}^k(x) = \sum_{\tilde{n}_{k-1} \sim p_{\theta_k}} \cdots \sum_{\tilde{n}_1 \sim p_{\theta_1}} [\nabla_w L((t_1 \circ t_{\tilde{n}_{k-1}} \cdots \circ t_{\tilde{n}_1})(x); w), \cdots , \tag{7}$$

$$\nabla_w L((t_{|\mathbb{T}|} \circ t_{\tilde{n}_{k-1}} \circ \cdots \circ t_{\tilde{n}_1})(x); w)]$$

where $\tilde{n}_{k-1} \sim p_{\theta_{k-1}}(n), \cdots, \tilde{n}_1 \sim p_{\theta_1}(n)$.

The average gradient at the $k^{th}$ layer can be estimated by the Monte Carlo method as:

$$\tilde{g}(x; \theta_k) = \sum_{\tilde{n}_k \sim p_{\theta_k}} \cdots \sum_{\tilde{n}_1 \sim p_{\theta_1}} \nabla_w L((t_{\tilde{n}_k} \circ \cdots \circ t_{\tilde{n}_1})(x); w). \tag{8}$$

Therefore, the reward at the $k^{th}$ layer is derived as:

$$\tilde{r}^k(x) = \left(\tilde{G}^k(x)\right)^T \left(\frac{v}{\|\tilde{g}_k(x; \theta_k)\|} - \frac{v^T \tilde{g}_k(x; \theta_k)}{\|\tilde{g}_k(x; \theta_k)\|^2} \cdot \frac{\tilde{g}_k(x; \theta_k)}{\|\tilde{g}_k(x; \theta_k)\|}\right). \tag{9}$$

To prevent the augmentation policy from overfitting, we regularize the optimization by avoiding optimizing towards the direction with high variance. Thus, we penalize the average reward with its standard deviation as

$$r^k = E_x\{\tilde{r}^k(x)\} - c \cdot \sqrt{E_x\{(\tilde{r}^k(x) - E_x\{\tilde{r}^k(x)\})^2\}}, \tag{10}$$

where we use 16 randomly sampled images to calculate the expectation. The hyperparameter $c$ controls the degree of regularization, which is set to $1.0$. With such regularization, we prevent the policy from converging to the transformations with high variance.

Therefore the parameters of policy $\mathcal{P}_k$ ($k \geq 2$) can be updated as:

$$\theta \leftarrow \theta_k + \eta \nabla_{\theta_k} \text{cosineSimilarity}(v, g(\theta_k)) \tag{11}$$

where

$$\nabla_\theta \text{cosineSimilarity}(v, g^k(x; \theta)) = \nabla_\theta p_{\theta_k} \cdot r^k. \tag{12}$$

## 4 EXPERIMENTS AND ANALYSIS

**Benchmarks and Baselines.** We evaluate the performance of DeepAA on three standard benchmarks: CIFAR-10, CIFAR-100, ImageNet, and compare it against a baseline based on standard augmentations (*i.e.*, flip left-righ, pad-and-crop for CIFAR-10/100, and Inception-style preprocesing (Szegedy et al., 2015) for ImageNet) as well as nine existing automatic augmentation methods including (1) AutoAugment (AA) (Cubuk et al., 2019), (2) PBA (Ho et al., 2019), (3) Fast AutoAugment (Fast AA) (Lim et al., 2019), (4) Faster AutoAugment (Hataya et al., 2020), (5) DADA (Li et al., 2020), (6) RandAugment (RA) (Cubuk et al., 2020), (7) UniformAugment (UA) (LingChen et al., 2020), (8) TrivialAugment (TA) (Müller & Hutter, 2021), and (9) Adversarial AutoAugment (AdvAA) (Zhang et al., 2019).

**Search Space.** We set up the operation set $\mathbb{O}$ to include 16 commonly used operations (identity, shear-x, shear-y, translate-x, translate-y, rotate, solarize, equalize, color, posterize, contrast, brightness, sharpness, autoContrast, invert, Cutout) as well as two operations (*i.e.*, flips and crop) that are used as the default operations in the aforementioned methods. Among the operations in $\mathbb{O}$, 11 operations are associated with magnitudes. We then discretize the range of magnitudes into 12 uniformly spaced levels and treat each operation with a discrete magnitude as an independent transformation. Therefore, the policy in each layer is a 139-dimensional categorical distribution corresponding to $|\mathbb{T}| = 139$ {operation, magnitude} pairs. The list of operations and the range of magnitudes in the standard augmentation space are summarized in Appendix A.

## 4.1 Performance on CIFAR-10 and CIFAR-100

**Policy Search.** Following (Cubuk et al., 2019), we conduct the augmentation policy search based on Wide-ResNet-40-2 (Zagoruyko & Komodakis, 2016). We first train the network on a subset of $4,000$ randomly selected samples from CIFAR-10. We then progressively update the policy network parameters $\theta_k$ $(k = 1, 2, \cdots, K)$ for 512 iterations for each of the $K$ augmentation layers. We use the Adam optimizer (Kingma & Ba, 2015) and set the learning rate to $0.025$ for policy updating.

**Policy Evaluation.** Using the publicly available repository of Fast AutoAugment (Lim et al., 2019), we evaluate the found augmentation policy on both CIFAR-10 and CIFAR-100 using Wide-ResNet-28-10 and Shake-Shake-2x96d models. The evaluation configurations are kept consistent with that of Fast AutoAugment.

**Results.** Table 1 reports the Top-1 test accuracy on CIFAR-10/100 for Wide-ResNet-28-10 and Shake-Shake-2x96d, respectively. The results of DeepAA are the average of four independent runs with different initializations. We also show the $95\%$ confidence interval of the mean accuracy. As shown, DeepAA achieves the best performance compared against previous works using the standard augmentation space. *Note that TA(Wide) uses a wider (stronger) augmentation space on this dataset.*

| | Baseline | AA | PBA | FastAA | FasterAA | DADA | RA | UA | TA(RA) | TA(Wide) [1] | DeepAA |
|---|---|---|---|---|---|---|---|---|---|---|---|
| **CIFAR-10** | | | | | | | | | | | |
| WRN-28-10 | 96.1 | 97.4 | 97.4 | 97.3 | 97.4 | 97.3 | 97.3 | 97.33 | 97.46 | 97.46 | **97.56** $\pm$ 0.14 |
| Shake-Shake (26 2x96d) | 97.1 | 98.0 | 98.0 | 98.0 | 98.0 | 98.0 | 98.0 | 98.1 | 98.05 | 98.21 | **98.11** $\pm$ 0.12 |
| **CIFAR-100** | | | | | | | | | | | |
| WRN-28-10 | 81.2 | 82.9 | 83.3 | 82.7 | 82.7 | 82.5 | 83.3 | 82.82 | 83.54 | 84.33 | **84.02** $\pm$ 0.18 |
| Shake-Shake (26 2x96d) | 82.9 | 85.7 | 84.7 | 85.1 | 85.0 | 84.7 | - | - | - | 86.19 | **85.19** $\pm$ 0.28 |

Table 1: Top-1 test accuracy on CIFAR-10/100 for Wide-ResNet-28-10 and Shake-Shake-2x96d. The results of DeepAA are averaged over four independent runs with different initializations. The $95\%$ confidence interval is denoted by $\pm$.

## 4.2 Performance on ImageNet

**Policy Search.** We conduct the augmentation policy search based on ResNet-18 (He et al., 2016). We first train the network on a subset of $200,000$ randomly selected samples from ImageNet for 30 epochs. We then use the same settings as in CIFAR-10 for updating the policy parameters.

**Policy Evaluation.** We evaluate the performance of the found augmentation policy on ResNet-50 and ResNet-200 based on the public repository of Fast AutoAugment (Lim et al., 2019). The parameters for training are the same as the ones of (Lim et al., 2019). In particular, we use step learning rate scheduler with a reduction factor of $0.1$, and we train and evaluate with images of size 224x224.

**Results.** The performance on ImageNet is presented in Table 2. As shown, DeepAA achieves the best performance compared with previous methods without the use of default augmentation pipeline. In particular, DeepAA performs better on larger models (*i.e.* ResNet-200), as the performance of DeepAA on ResNet-200 is the best within the $95\%$ confidence interval. *Note that while we train DeepAA using the image resolution (224×224), we report the best results of RA and TA, which are trained with a larger image resolution (244×224) on this dataset.*

| | Baseline | AA | Fast AA | Faster AA | DADA | RA | UA | TA(RA)[1] | TA(Wide)[2] | DeepAA |
|---|---|---|---|---|---|---|---|---|---|---|
| ResNet-50 | 76.3 | 77.6 | 77.6 | 76.5 | 77.5 | 77.6 | 77.63 | 77.85 | 78.07 | **78.30** $\pm$ 0.14 |
| ResNet-200 | 78.5 | 80.0 | 80.6 | - | - | - | 80.4 | - | - | **81.32** $\pm$ 0.17 |

Table 2: Top-1 test accuracy (%) on ImageNet for ResNet-50 and ResNet-200. The results of DeepAA are averaged over four independent runs with different initializations. The $95\%$ confidence interval is denoted by $\pm$.

## 4.3 Performance with Batch Augmentation

Batch Augmentation (BA) is a technique that draws multiple augmented instances of the same sample in one mini-batch. It has been shown to be able to improve the generalization performance of the

---

[1]On CIFAR-10/100, TA (Wide) uses a wider (stronger) augmentation space, while the other methods including TA (RA) uses the standard augmentation space.

network (Berman et al., 2019; Hoffer et al., 2020). AdvAA (Zhang et al., 2019) directly searches for the augmentation policy under the BA setting whereas for TA and DeepAA, we apply BA with the same augmentation policy used in Table 1. Note that since the performance of BA is sensitive to the hyperparameters (Fort et al., 2021), we have conducted a grid search on the hyperparameters of both TA and DeepAA (details are included in Appendix D). As shown in Table 3, after tuning the hyperparameters, the performance of TA (Wide) using BA is already better than the reported performance in the original paper. The performance of DeepAA with BA outperforms that of both AdvAA and TA (Wide) with BA.

|  | **AdvAA** | **TA(Wide)** (original paper) | **TA(Wide)** (ours) | **DeepAA** |
|---|---|---|---|---|
| CIFAR-10 | $98.1 \pm 0.15$ | $98.04 \pm 0.06$ | $98.06 \pm 0.23$ | $\mathbf{98.21 \pm 0.14}$ |
| CIFAR-100 | $84.51 \pm 0.18$ | $84.62 \pm 0.14$ | $85.40 \pm 0.15$ | $\mathbf{85.61 \pm 0.17}$ |

Table 3: Top-1 test accuracy (%) on CIFAR-10/100 dataset with WRN-28-10 with Batch Augmentation (BA), where eight augmented instances were drawn for each image. The results of DeepAA are averaged over four independent runs with different initializations. The 95% confidence interval is denoted by $\pm$.

### 4.4 UNDERSTANDING DEEPAA

**Effectiveness of Gradient Matching.** One uniqueness of DeepAA is the regularized gradient matching objective. To examine its effectiveness, we remove the impact coming from multiple augmentation layers, and only conduct search for a single layer of augmentation policy. When evaluating the searched policy, we apply the default augmentation in addition to the searched policy. We refer to this variant as DeepAA-simple. Figure 2 compares the Top-1 test accuracy on ImageNet using ResNet-50 between DeepAA-simple, DeepAA, and other automatic augmentation methods. While there is 0.22% performance drop compared to DeepAA,

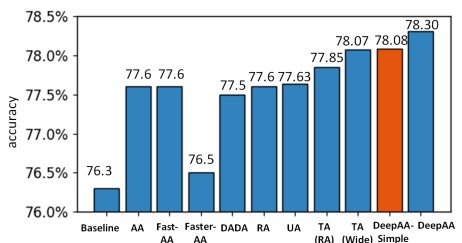

Figure 2: Top-1 test accuracy (%) on ImageNet of DeepAA-simple, DeepAA, and other automatic augmentation methods on ResNet-50.

*with a single augmentation layer*, DeepAA-simple still outperforms other methods and is able to achieve similar performance compared to TA (Wide) but with a standard augmentation space and trains on a smaller image size (224×224 vs 244×224).

**Policy Search Cost.** Table 4 compares the policy search time on CIFAR-10/100 and ImageNet in GPU hours. DeepAA has comparable search time as PBA, Fast AA, and RA, but is slower than Faster AA and DADA. Note that Faster AA and DADA relax the discrete search space to a continuous one similar to DARTS (Liu et al., 2018). While such relaxation leads to shorter searching time, it inevitably introduces a discrepancy between the true and relaxed augmentation spaces.

| Dataset | AA | PBA | Fast AA | Faster AA | DADA | RA | DeepAA |
|---|---|---|---|---|---|---|---|
| CIFAR-10/100 | 5000 | 5 | 3.5 | 0.23 | 0.1 | 25 | 9 |
| ImageNet | 15000 | - | 450 | 2.3 | 1.3 | 5000 | 96 |

Table 4: Policy search time on CIFAR-10/100 and ImageNet in GPU hours.

**Impact of the Number of Augmentation Layers.** Another uniqueness of DeepAA is its multi-layer search space that can *go beyond* two layers which existing automatic augmentation methods were designed upon. We examine the impact of the number of augmentation layers on the performance of DeepAA. Table 5 and Table 6 show the performance on CIFAR-10/100 and ImageNet respectively with increasing number of augmentation layers. As shown, for CIFAR-10/100, the performance gradually improves when more augmentation layers are added until we reach five layers. The performance does not improve when the sixth layer is added. For ImageNet, we have similar

---

[1]TA (RA) achieves 77.55% top-1 accuracy with image resolution 224×224.

[2]TA (Wide) achieves 77.97% top-1 accuracy with image resolution 224×224.

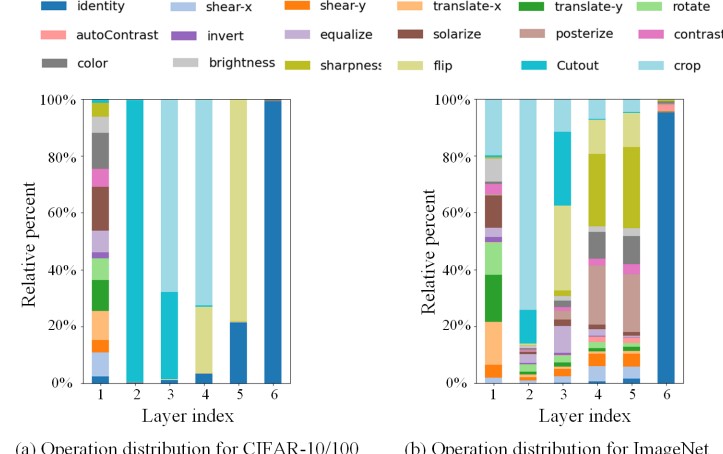

(a) Operation distribution for CIFAR-10/100  (b) Operation distribution for ImageNet

Figure 3: The distribution of operations at each layer of the policy for CIFAR-10/100 and ImageNet. The probability of each operation is summed up over all 12 discrete intensity levels (see Appendix B and C) of the corresponding transformation.

observation where the performance stops improving when more than five augmentation layers are included.

|  | 1 layer | 2 layers | 3 layers | 4 layers | 5 layers | 6 layers |
|---|---|---|---|---|---|---|
| CIFAR-10 | 96.3 ± 0.21 | 96.6 ± 0.18 | 96.9 ± 0.12 | 97.4 ± 0.14 | **97.56 ± 0.14** | **97.6 ± 0.12** |
| CIFAR-100 | 80.9 ± 0.31 | 81.7 ± 0.24 | 82.2 ± 0.21 | 83.7 ± 0.24 | **84.02 ± 0.18** | **84.0 ± 0.19** |

Table 5: Top-1 test accuracy of DeepAA on CIFAR-10/100 for different numbers of augmentation layers. The results are averaged over 4 independent runs with different initializations with the 95% confidence interval denoted by ±.

|  | 1 layer | 3 layers | 5 layers | 7 layers |
|---|---|---|---|---|
| ImageNet | 75.27 ± 0.19 | 78.18 ± 0.22 | **78.30 ± 0.14** | **78.30 ± 0.14** |

Table 6: Top-1 test accuracy of DeepAA on ImageNet with ResNet-50 for different numbers of augmentation layers. The results are averaged over 4 independent runs w/ different initializations with the 95% confidence interval denoted by ±.

Figure 3 illustrates the distributions of operations in the policy for CIFAR-10/100 and ImageNet respectively. As shown in Figure 3(a), the augmentation of CIFAR-10/100 converges to *identity* transformation at the sixth augmentation layer, which is a natural indication of the end of the augmentation pipeline. We have similar observation in Figure 3(b) for ImageNet, where the *identity* transformation dominates in the sixth augmentation layer. These observations match our results listed in Table 5 and Table 6. We also include the distribution of the magnitude within each operation for CIFAR-10/100 and ImageNet in Appendix B and Appendix C.

**Validity of Optimizing Gradient Matching with Regularization.** To evaluate the validity of optimizing gradient matching with regularization, we designed a search-free baseline named "DeepTA". In DeepTA, we stack multiple layers of TA on the same augmentation space of DeepAA without using default augmentations. As stated in Eq.(10) and Eq.(12), we explicitly optimize the gradient similarities with the average reward minus its standard deviation. The first term – the average reward $E_x\{\tilde{r}^k(x)\}$ – *encourages the direction of high cosine similarity*. The second term – the standard deviation of the reward $\sqrt{E_x\{(\tilde{r}^k(x) - E_x\{\tilde{r}^k(x)\})^2\}}$ – acts as a regularization that *penalizes the direction with high variance*. These two terms jointly maximize the gradient similarity along the direction with low variance. To illustrate the optimization trajectory, we design two metrics that are closely related to the two terms in Eq.(10): the mean value, and the standard deviation of the improvement of gradient similarity. The improvement of gradient similarity is obtained by subtracting the cosine similarity of the original image batch from that of the augmented batch. In our experiment, the mean and standard deviation of the gradient similarity improvement are calculated over 256 independently sampled original images.

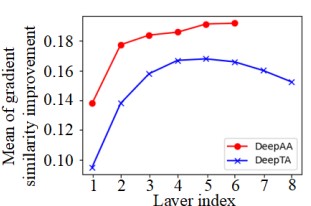 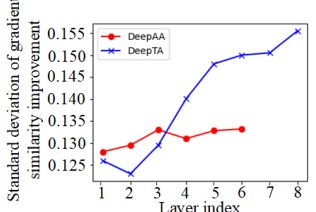 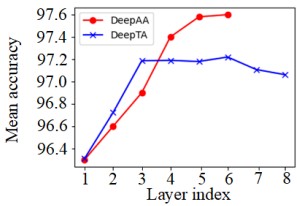

(a) Mean of the gradient similarity improvement (b) Standard deviation of the gradient similarity improvement (c) Mean accuracy over different augmentation depth

Figure 4: Illustration of the search trajectory of DeepAA in comparison with DeepTA on CIFAR-10.

As shown in Figure 4(a), the cosine similarity of DeepTA reaches the peak at the fifth layer, and stacking more layers decreases the cosine similarity. In contrast, for DeepAA, the cosine similarity increases consistently until it converges to *identity* transformation at the sixth layer. In Figure 4(b), the standard deviation of DeepTA significantly increases when stacking more layers. In contrast, in DeepAA, as we optimize the gradient similarity along the direction of low variance, the standard deviation of DeepAA does not grow as fast as DeepTA. In Figure 4(c), both DeepAA and DeepTA reach peak performance at the sixth layer, but DeepAA achieves better accuracy compared against DeepTA. Therefore, we empirically show that DeepAA effectively scales up the augmentation depth by increasing cosine similarity along the direction with low variance, leading to better results.

**Comparison with Other Policies.** In Figure 7 in Appendix E, we compare the policy of DeepAA with the policy found by other data augmentation search methods including AA, FastAA and DADA. We have three interesting observations:

- AA, FastAA and DADA assign high probability (over 1.0) on flip, Cutout and crop, as those transformations are hand-picked and applied by default. DeepAA finds a similar pattern that assigns high probability on flip, Cutout and crop.

- Unlike AA, which mainly focused on color transformations, DeepAA has high probability over both spatial and color transformations.

- FastAA has evenly distributed magnitudes, while DADA has low magnitudes (common issues in DARTS-like method). Interestingly, DeepAA assigns high probability to the stronger magnitudes.

## 5 CONCLUSION

In this work, we present Deep AutoAugment (DeepAA), a multi-layer data augmentation search method that finds deep data augmentation policy without using any hand-picked default transformations. We formulate data augmentation search as a regularized gradient matching problem, which maximizes the gradient similarity between augmented data and original data along the direction with low variance. Our experimental results show that DeepAA achieves strong performance without using default augmentations, indicating that regularized gradient matching is an effective search method for data augmentation policies.

**Reproducibility Statement:** We have described our experiment settings in great details. The evaluation of the found data augmentation policy is based the public repository of Fast AutoAugment. We believe that our results can be readily reproduced.

## ACKNOWLEDGEMENT

We thank Yi Zhu, Hang Zhang, Haichen Shen, Mu Li, and Alexander Smola for their help with this work. This work was partially supported by NSF Award PFI:BIC-1632051 and Amazon AWS Machine Learning Research Award.

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

# A  A LIST OF STANDARD AUGMENTATION SPACE

| Operation | Magnitude |
|---|---|
| Identity | - |
| ShearX | [-0.3, 0.3] |
| ShearY | [-0.3, 0.3] |
| TranslateX | [-0.45, 0.45] |
| TranslateY | [-0.45, 0.45] |
| Rotate | [-30, 30] |
| AutoContrast | - |
| Invert | - |
| Equalize | - |
| Solarize | [0, 256] |
| Posterize | [4, 8] |
| Contrast | [0.1, 1.9] |
| Color | [0.1, 1.9] |
| Brightness | [0.1, 1.9] |
| Sharpness | [0.1, 1.9] |
| Flips | - |
| Cutout | 16 (60) |
| Crop | - |

Table 7: List of operations in the search space and the corresponding range of magnitudes in the standard augmentation space. Note that some operations do not use magnitude parameters. We add flip and crop to the search space which were found in the default augmentation pipeline in previous works. Flips operates by randomly flipping the images with $50\%$ probability. In line with previous works, crop denotes pad-and-crop and resize-and-crop transforms for CIFAR10/100 and ImageNet respectively. We set Cutout magnitude to 16 for CIFAR10/100 dataset to be the same as the Cutout in the default augmentation pipeline. We set Cutout magnitude to 60 pixels for ImageNet which is the upper limit of the magnitude used in AA (Cubuk et al., 2019).

## B THE DISTRIBUTION OF MAGNITUDES FOR CIFAR-10/100

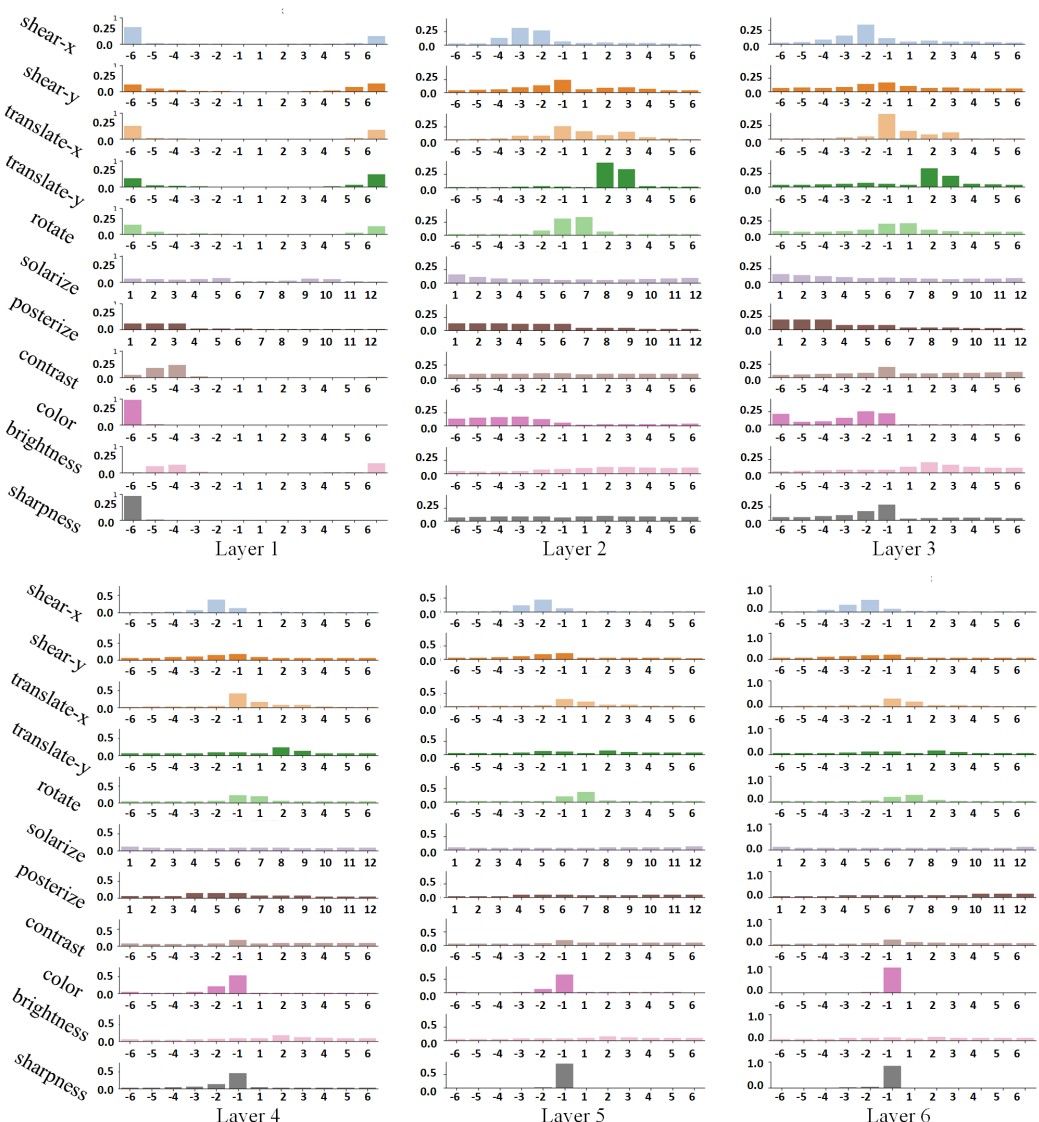

Figure 5: The distribution of discrete magnitudes of each augmentation transformation in each layer of the policy for CIFAR-10/100. The x-axis represents the discrete magnitudes and the y-axis represents the probability. The magnitude is discretized to 12 levels with each transformation having its own range. A large absolute value of the magnitude corresponds to high transformation intensity. Note that we do not show identity, autoContrast, invert, equalize, flips, Cutout and crop because they do not have intensity parameters.

## C  THE DISTRIBUTION OF MAGNITUDES FOR IMAGENET

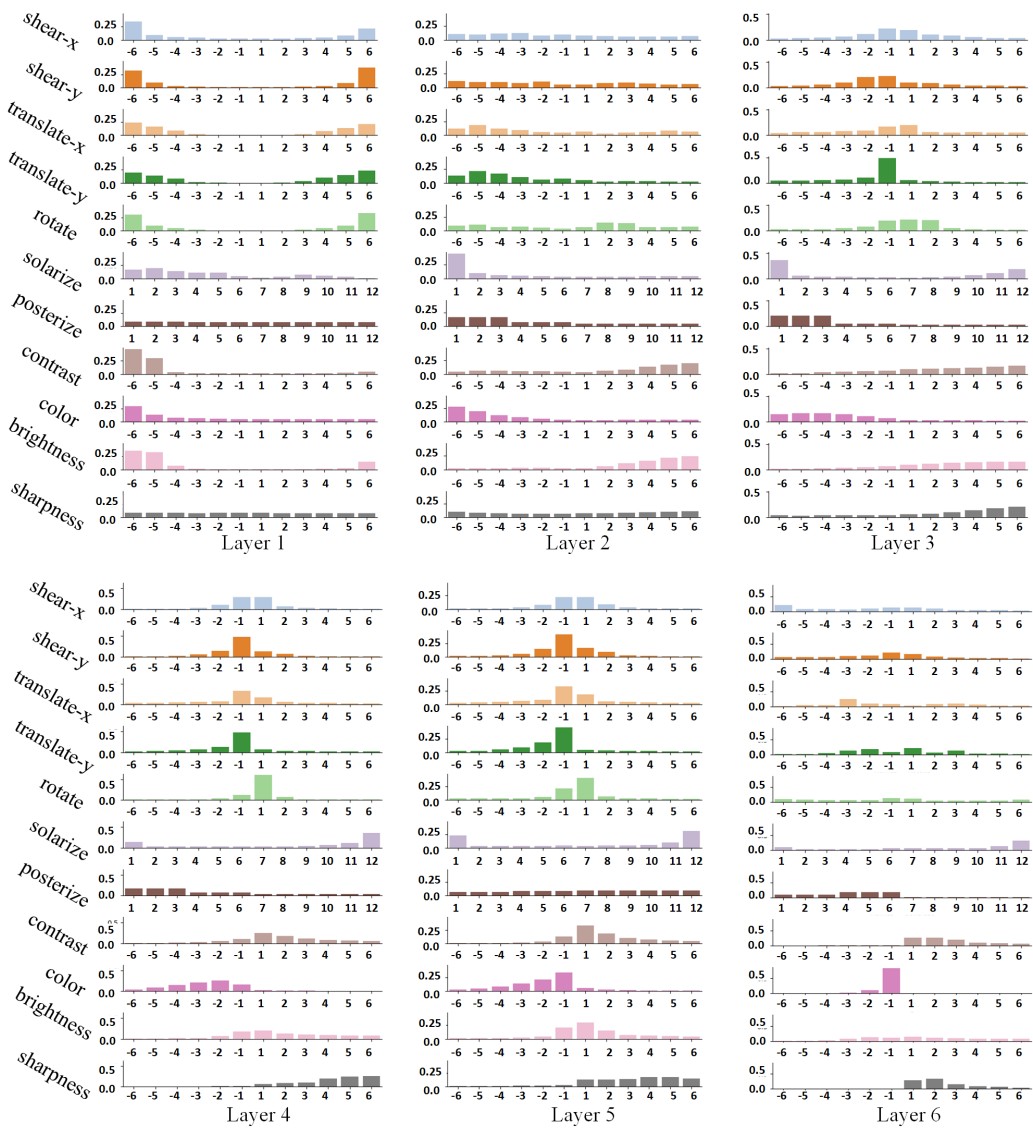

Figure 6: The distribution of discrete magnitudes of each augmentation transformation in each layer of the policy for ImageNet. The x-axis represents the discrete magnitudes and the y-axis represents the probability. The magnitude is discretized to 12 levels with each transformation having its own range. A large absolute value of the magnitude corresponds to high transformation intensity. Note that we do not show identity, autoContrast, invert, equalize, flips, Cutout and crop because they do not have intensity parameters.

## D  HYPERPARAMETERS FOR BATCH AUGMENTATION

The performance of BA is sensitive to the training settings (Fort et al., 2021; Wightman et al., 2021). Therefore, we conduct a grid search on the learning rate, weight decay and number of epochs for TA and DeepAA with Batch Augmentation. The best found parameters are summarized in Table 8 in Appendix. We did not tune the hyperparameters of AdvAA (Zhang et al., 2019) since AdvAA claims to be adaptive to the training process.

| Dataset | Augmentation | Model | Batch Size | Learning Rate | Weight Decay | Epoch |
|---|---|---|---|---|---|---|
| CIFAR-10 | TA (Wide) | WRN-28-10 | $128 \times 8$ | 0.2 | 0.0005 | 100 |
| | DeepAA | WRN-28-10 | $128 \times 8$ | 0.2 | 0.001 | 100 |
| CIFAR-100 | TA (Wide) | WRN-28-10 | $128 \times 8$ | 0.4 | 0.0005 | 35 |
| | DeepAA | WRN-28-10 | $128 \times 8$ | 0.4 | 0.0005 | 35 |

Table 8: Model hyperparameters of Batch Augmentation on CIFAR10/100 for TA (Wide) and DeepAA. Learning rate, weight decay and number of epochs are found via grid search.

# E    COMPARISON OF DATA AUGMENTATION POLICY

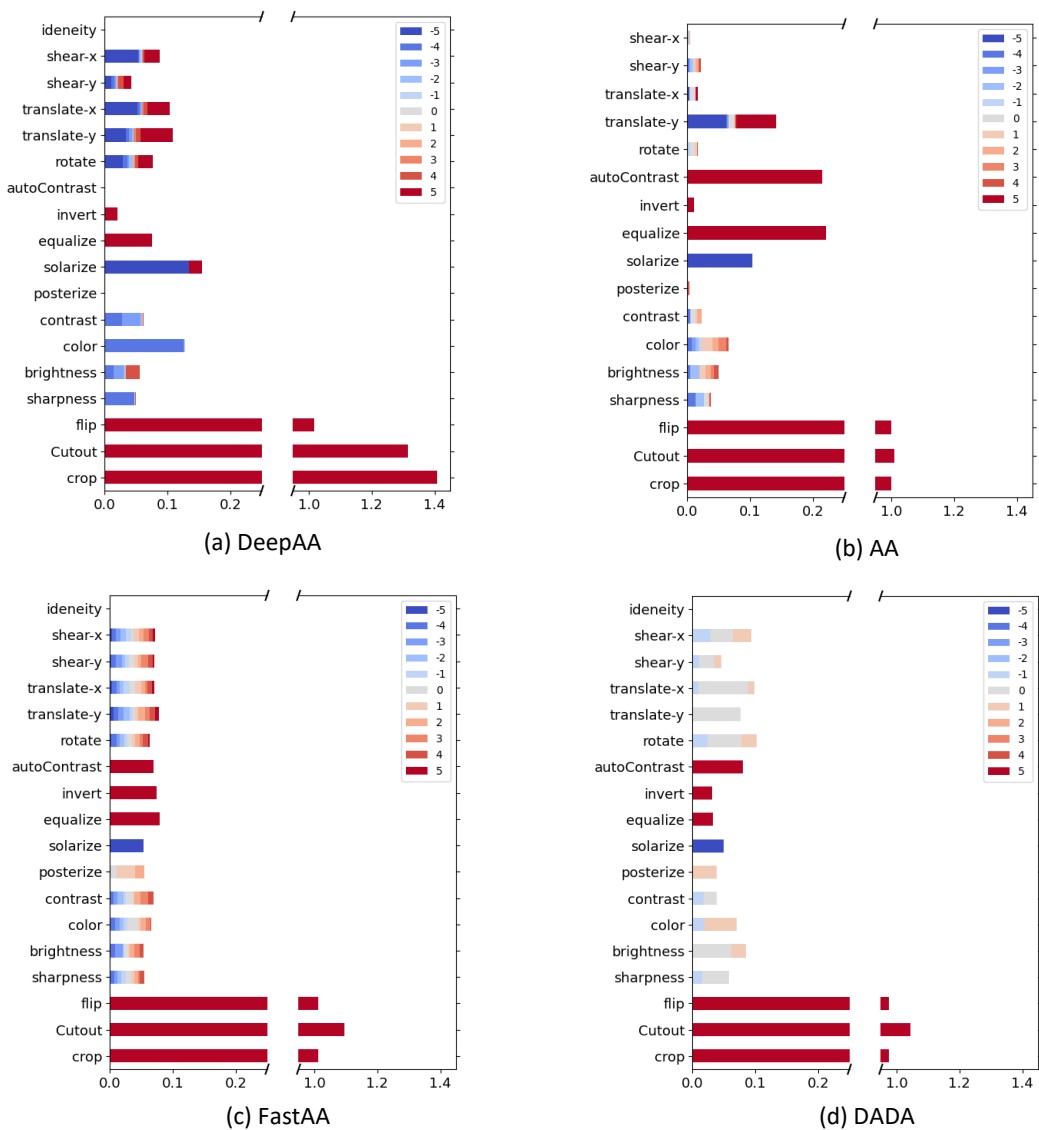

Figure 7: Comparison of the policy of DeepAA and some publicly available augmentaiotn policy found by other methods including AA, FastAA and DADA on CIFAR-10. Since the compared methods have varied numbers of augmentation layers, we cumulate the probability of each operation over all the augmentation layers. Thus, the cumulative probability can be larger than 1. For AA, Fast AA and DADA, we add additional 1.0 probability to flip, Cutout and Crop, since they are applied by default. In addition, we normalize the magnitude to the range [-5, 5], and use color to distinguish different magnitudes.

