# OpenReview forum: "Deep AutoAugment"
_ICLR.cc/2022/Conference — ICLR 2022 Poster_

### Official Review · Reviewer_BGUU · 2021-10-31

**Correctness:** 2
**Technical Novelty And Significance:** 3
**Empirical Novelty And Significance:** 2
**Recommendation:** 5
**Confidence:** 5

**Main Review:**

Strengths
- Getting rid of default augmentations is an interesting step
- Approach to get around the need of RL in previous work
    - Problem: Little misalignment with final objective, 1 sample vs full mean
- Nice visualization of the policy choice of DeepAA
- Repeated experiments with confidence intervals

Not Clear
- Do you optimize all augmentation layers at once or in separate optimization steps?

Weaknesses
- The statement about SOTA in the abstract is _not true_. The SOTA for C-10 with WRN-28-10 is at least 84.33 as reported in TrivialAugment, which is stronger than the 84.02 of this method. Similarly, TrivialAugment is forgotten in the comparison for ImageNet, even though here Deep AA is stronger but not as much as stated.
- I believe stating to use 139 augmentations is a little misleading, as you treat the same augmentation with different strengths as different augmentations
- Another experiment on the relatively standard SVHN datasets would be interesting, as this has very different images.
- Claim of not using default augmentations is weakened by the fact that the search space is changed for datasets that require different default augmentations
- Comparison to TrivialAugment is a little unfair, as TrivialAugment’s standard setting is stronger than the results presented, and TA actually includes a run with CIFAR-100 and ShakeShake
- Missing true baseline. I would be more sure of the method's performance if there was a simple baseline implemented on the same search space and in your codebase, either UA or TA: This could actually work out very well for you, too, if you include another baseline of a trivial baseline without standard augmentations, too.
- A baseline for Table 6 would be good to answer whether the cosine similarity increase with any augmentation application? That could be done by applying RA with different numbers of layers or applying TA multiple times.

Very minor: there are some typos here and there.


**Summary Of The Paper:**

The paper proposes a novel automatic augmentation method. That is a method that learns based on a dataset, how to augment that dataset during training of a classifier.

The proposed method works by matching gradients based on cosine similarity on fully trained networks and uses this approach to build augmentation policies with multiple augmentations applied after one another. Similar to the recent usage of very deep RandAugment for EfficientNetV2s for example.

**Summary Of The Review:**

Overall, I believe the proposed method is interesting: it builds interesting policies in an intuitive way and requires less per-dataset adaptions than previous work.

I do believe, though, that the results simply are not strong enough for publication and a few comparisons in the paper are unfair to baselines. It might be possible to improve upon this method in some ways though, for example with Batch Augmentation ("Augment Your Batch:.." by Hoffer et al.), as this is closer to \theta’s objective.

---

> ### Author Response · Authors · 2021-11-19
> **Response to reviewer BGUU (6/6)**
>
> > (6/6) Do you optimize all augmentation layers at once or in separate optimization steps?
>
> We optimize the augmentation policy from the first layer to the last layer in separate optimization steps, where the k-th layer is optimized based on the data distribution augmented by previous (k-1) layers.

---

> ### Author Response · Authors · 2021-11-19
> **Response to reviewer BGUU (5/6)**
>
> > (5/6) Stating to use 139 augmentations is a little misleading.
>
> We say that we have |T|=139 augmentation because the distribution p_{\theta} is a 139-dimensional categorical distribution where each dimension corresponds to an image operation with a specific magnitude. Moreover, by visual inspection, there exist tangible differences between images transformed by the same operations with different magnitudes. We will change the wording to “We then discretize the range of magnitudes into 12 uniformly spaced levels and treat each operation with a discrete magnitude as an independent transformation. Therefore, the policy in each layer is a 139-dimensional Categorical distribution corresponding to |T|=139 operation and magnitude pairs.”

---

> ### Author Response · Authors · 2021-11-19
> **Response to reviewer BGUU (4/6)**
>
> > (4/6) Claim of not using default augmentations is weakened by the fact that the search space is changed for datasets that require different default augmentations
>
> We change the augmentation space because we want to be consistent with the augmentation space in previous works for a fair comparison. The only two operations changed are Cutout and Crop. For CIFAR-10/100 we set Cutout and Crop to 16 pixels cutout and pad-and-crop, which are exactly the same as the implementation in previous works. We use resize-and-crop for ImageNet which is also exactly the same as the implementation in previous works. We set Cutout to 60 pixels for ImageNet as the image resolution is much higher, which is also the upper limit as implemented in AA.

---

> ### Author Response · Authors · 2021-11-19
> **Response to reviewer BGUU (3/6)**
>
> > (3/6) Improved Performance with Batch Augmentation (BA).
>
> We followed the comment to conduct an experiment on combining DeepAA with batch augmentation (BA). In the following table, we compare the results of DeepAA + BA, TA (Wide) + BA, and AdvAA, which uses BA as its main technique. Since BA is extremely sensitive to the hyperparameters [1,2,3], we therefore independently did a grid search on the learning rate, weight decay and the number of epochs for both TA (Wide) + BA and DeepAA + BA. We did not tune the hyperparameters of AdvAA as it claims to be adaptive to the training process.
>
> |                    |     AdvAA    | TA (Wide) + BA (original paper) | TA (Wide) + BA (ours) |  DeepAA + BA  |
> |--------------------|:------------:|:-------------------------------:|:---------------------:|:-------------:|
> |  CIFAR10 WRN-28-10 |  98.1 ± 0.15 |           98.04 ± 0.06          |      98.06 ± 0.23     |  **98.21 ± 0.14** |
> | CIFAR100 WRN-28-10 | 84.51 ± 0.18 |           84.62 ± 0.14          |      85.40 ± 0.15     | **85.61 ± 0.17**  |
> | |  |   |   |  |
>
> We have two observations.
> 1.  After tuning the hyperparameters, the performance of TA (Wide) + BA is better than the reported performance in the original TA paper.
> 2. DeepAA + BA indeed enhances the performance of DeepAA, and it outperforms TA (Wide) + BA after hyperparameter tuning.
>
> The best hyperparameters we found are summarized in the table below:
>
> | Dataset   | Augmentation | Model   | Batch Size | LR  | WD     | Epoch |
> |-----------|--------------|---------|------------|-----|--------|-------|
> | CIFAR-10  | TA (Wide)    | WRN-28-10 | 128 * 8    | 0.2 | 0.0005 | 100   |
> | CIFAR-10  | DeepAA       | WRN-28-10 | 128 * 8    | 0.2 | 0.001  | 100   |
> | CIFAR-100 | TA (Wide)    | WRN-28-10 | 128 * 8    | 0.4 | 0.0005 | 35    |
> | CIFAR-100 | DeepAA       | WRN-28-10 | 128 * 8    | 0.4 | 0.0005 | 35    |
>
> [1] Zhang, Xinyu, et al. "Adversarial AutoAugment." International Conference on Learning Representations. 2019.
>
> [2] Fort, Stanislav, et al. "Drawing Multiple Augmentation Samples Per Image During Training Efficiently Decreases Test Error." arXiv preprint arXiv:2105.13343 (2021).
>
> [3] Wightman, Ross, Hugo Touvron, and Hervé Jégou. "ResNet strikes back: An improved training procedure in timm." arXiv preprint arXiv:2110.00476 (2021).

---

> > ### Comment · Reviewer_BGUU · 2021-11-25
> > **Details**
> >
> > Very interesting experiment. I did not expect you would push out experiments on this! A little sad of course that DeepAA did not do even better, as I suggested. Sorry for that!

---

> > > ### Author Response · Authors · 2021-11-26
> > > **Response to reviewer BGUU**
> > >
> > > As we showed in our response above, DeepAA+BA indeed outperforms both AdvAA and TA (Wide)+BA. If possible, could the reviewer help clarify the comment a little bit? Thanks a lot.

---

> ### Author Response · Authors · 2021-11-19
> **Response to reviewer BGUU (2/6)**
>
> > (2/6) Missing true baseline.
>
> We followed the comment to design a true baseline named “**DeepTA**”. In DeepTA, we stack multiple layers of TA on the RA augmentation space without default augmentations. In the following table, we compare the results of DeepAA and DeepTA across layers.
>
> |                    |        |   1 Layer   |   2 Layers  |   3 Layers  |   4 Layers   |   5 Layers  |   6 Layers   |    7 Layers   |   8 Layers   |
> |--------------------|--------|:-----------:|:-----------:|:-----------:|:------------:|:-----------:|:------------:|:-------------:|:------------:|
> | CIFAR10 WRN-28-10  | DeepTA |  96.3±0.14  |  96.72±0.16 |  97.18±0.07 |  97.19±0.10  |  97.18±0.17 | **97.22±0.13** | 97.11±0.15 |  97.06±0.1 |
> | CIFAR10 WRN-28-10  | DeepAA |  96.3±0.21  |  96.6±0.18  |  96.9±0.12  |   97.4±0.14  |  **97.56±0.14** |   **97.6±0.12**  |       -       |       -      |
> | CIFAR100 WRN-28-10 | DeepTA | 80.5±0.17 | 81.94±0.25 | 82.32±0.29 | 82.56±0.26 | 82.60±0.16 |  82.67±0.22 | **82.89±0.21** | 82.62±0.36 |
> | CIFAR100 WRN-28-10 | DeepAA |  80.9±0.31  |  81.7±0.24  |  82.2±0.21  |   83.7±0.24  |  **84.02±0.18** |   **84.0±0.19**  |       -       |       -      |
> | | | | | |   |  |   |          |        |
>
> We have two observations.
>
> 1. The best-performing augmentation policy found by DeepAA consists of 5 layers for both CIFAR-10 and CIFAR-100. The best-performing augmentation policy found by DeepTA is 6 layers for CIFAR-10 and 7 layers for CIFAR-100. As shown, the best-performing augmentation policy found by DeepAA outperforms the one found by DeepTA.
>
> 2. For DeepAA, at layer 6, 7, and 8, the augmentation operators converge to identity transforms; thus the accuracies converge too. In contrast, for DeepTA, our results show that when the number of layers goes beyond 6 in CIFAR-10 and 7 in CIFAR-100, the accuracy starts to drop.

---

> > ### Comment · Reviewer_BGUU · 2021-11-25
> > **Details**
> >
> > Thanks for doing this experiment. Very interesting. One remark, though: Is this on a lower budget? In my understanding DeepTA with 1 layer should be the exact same as TA (RA), so should have something like a 97.5% acc with WRN28-10 on C-10.

---

> > > ### Author Response · Authors · 2021-11-26
> > > **Response to reviewer BGUU**
> > >
> > > In DeepTA (1 layer), the augmentation space is RA’s augmentation space, with flip, crop and Cutout included in the augmentation space. In contrast, TA (RA) (1 layer) is RA’s augmentation space, with flip, crop and Cutout concatenated as an additional 3-layer default augmentation.

---

> ### Author Response · Authors · 2021-11-19
> **Response to reviewer BGUU (1/6)**
>
> We thank the reviewer for the thoughtful review. Here are our responses.
> > (1/6) The statement about SOTA is not true.
>
> The results reported in our submission are from version 1 (v1) of TrivialAugment (TA) on arxiv. Those results were further improved in version 2 (v2) of TA on arxiv. The primary difference between v1 and v2 is the augmentation space: v1 uses the augmentation space of RandAugment (RA) (denoted as TA (RA)) while v2 uses a larger (stronger) augmentation space (denoted as TA (Wide)).
>
> In our submission, we selected the same augmentation space used in TA (RA), RandAugment (RA), Fast AutoAugment (FastAA), AutoAugment (AA), and DADA for the purpose of fair comparison.
>
> In the following table, we compare the results of TA (RA), TA (Wide), and DeepAA.
>
> |                       | TA (RA)                | TA (Wide)                                     | DeepAA                 |
> |-----------------------|------------------------|-----------------------------------------------|------------------------|
> | CIFAR-10 WRN-28-10    | 97.46 ± 0.09           |                  97.46 ± 0.06                 |      **97.56 ± 0.14**     |
> | CIFAR-10 Shake-Shake  | 98.05  ± 0.05          |                  **98.21 ± 0.06**                 |      98.11  ± 0.12     |
> | CIFAR-100 WRN-28-10   | 83.54 ± 0.12           |                  **84.33 ± 0.17**                 |      84.02 ± 0.18      |
> | CIFAR-100 Shake-Shake | N/A                    |                 **86.19  ±  0.15**                |      85.19 ± 0.28      |
> | ImageNet ResNet-50    | 77.85 ± 0.15 (244x224) | 78.07 ± 0.27 (244x224) 77.97 ± 0.21 (224x224) | **78.30 ± 0.14** (224x224) |
> | | | | |
>
> We have two observations.
> 1.  Compared to TA (RA), in the same augmentation space, DeepAA is 0.1%, 0.48% and 0.45% higher on CIFAR-10, CIFAR-100, and ImageNet. We argue that such improvements are not trivial.
> 2.  Compared to TA (Wide), TA (Wide) achieved SOTA on CIFAR-10 (Shake-Shake), CIFAR-100 (WRN-28-10), and CIFAR-100 (Shake-Shake) but with a larger (stronger) augmentation space. However, DeepAA outperforms TA (Wide) on ImageNet by 0.23% despite DeepAA uses a smaller (weaker) augmentation space and trains on a smaller image size (224x224 vs 244x224).

---

> ### Comment · Reviewer_BGUU · 2021-11-25
> **Strong Rebuttal**
>
> Thanks for adding these many experiments. The experiments of your rebuttal did show me that your method does seem to outperform baselines, but only slightly. Since the method is relatively complex, but the absolute improvements are not very strong, I am a little scared that it will be hard to build upon this paper. That is why I still tend towards the reject side, even though you improved the paper a lot in my eyes within this rebuttal.

---

> > ### Author Response · Authors · 2021-11-26
> > **Response to reviewer BGUU**
> >
> > We thank the reviewer for taking time to read our response.
> >
> > Compared to existing approaches, our method provides a new perspective on tackling the automatic augmentation policy search, which we hope could inspire more research work in this new direction.
> >
> > We will make our codes clean and well commented. We will also write a document to guide other researchers to use our codes, reproduce our results, and build upon our work with ease.

---

### Official Review · Reviewer_9qhP · 2021-11-02

**Correctness:** 4
**Technical Novelty And Significance:** 3
**Empirical Novelty And Significance:** 3
**Recommendation:** 8
**Confidence:** 4

**Main Review:**

Strengths:
+ Relaxing the strong human assumption on data augmentation for specific data is novel and well motivated.
+ The proposal of using cosine similarity between the augmented training data and validation data as a reward signal to optimize the sampling probability of the corresponding data augmentation transformations is technically sound.
+ Table 1, 2 shows its better performance than SOTA DAS methods. Table 4, 5 further shows its increasing performance given an increasing number of layers and its stable performance given multiple runs.
+ Figure 3 shows in the top (deep) layers the augmentations are not dataset-specific and therefore justify its motivation.

Weakness:
- Isn’t the optimal similarity between train/validation to transform validation data to training data? Could the author[s] provide some intuition on why using gradient matching for the search. If it will cause overfitting issues on training data?
- Fig.3, 4 and 5 show different operation selections at different stages for DeepAA. Could the author[s] show the difference between previous works?
- Since this work is to advocate less human-constraints on operator selections, it would make this work better if the author[s] can show experiments on data-diversity cases where there is no prior knowledge about the data transformation. Showing extra results on some transfer learning experiments on different datasets will be highly appreciated.

**Summary Of The Paper:**

This paper provides an interesting finding that instead of injecting experts prior to hand-pick augmentations at the top layers, the entire search can be formulated end-to-end in a multi-layer (deep) way without hand-picked transformation optimized with gradient matching. The independent sampling at each layer with estimated mean gradient of training data is used to stabilize the training and make the optimization process more efficient. Empirical results on CIFAR-10/100 and ImageNet outperform previous best performing DAS methods.

**Summary Of The Review:**

Nice motivation and an interesting approach. Results on standard benchmarks are strong compared to other DAS methods. Experiments can be further expanded to different datasets to show the effectiveness of the proposed method. Can also add more comparisons to the previous methods in Fig. 3/4/5.

---

> ### Author Response · Authors · 2021-11-20
> **Response to reviewer 9qhP (2/2)**
>
> > (2/2) Fig.3, 4 and 5 show different operation selections at different stages for DeepAA. Could the author[s] show the difference between previous works?
>
> We compare the augmentation policy found by DeepAA with the policies found by other methods including AA, Fast AA, and DADA in Figure which can be accessed via [the anonymous link](https://figshare.com/s/44176c3f7732f3924004).
> Since the compared methods have varied numbers of augmentation layers, we cumulate the probability of each operation over all the augmentation layers. Thus, the cumulative probability can be larger than 1. For AA, Fast AA and DADA, we add additional 1.0 probability to flip, Cutout and Crop, since they are applied by default. In addition, we round the magnitude to the nearest integer in the range [-5, 5], and use color to distinguish different magnitudes.
>
> We have three observations:
> 1. AA, FastAA and DADA assign high probability (over 1.0) on flip, Cutout and crop, as those transformations are hand-picked and applied by default. DeepAA finds a similar pattern that assigns high probability on flip, Cutout and crop.
> 2. Unlike AA which mainly focused on color transformations, DeepAA has high probability over both spatial and color transformations.
> 3. FastAA has evenly distributed magnitudes, while DADA has low magnitudes. DeepAA, in contrast, assigns high probability on the stronger magnitudes.

---

> ### Author Response · Authors · 2021-11-20
> **Response to reviewer 9qhP (1/2)**
>
> >  Isn’t the optimal similarity between train/validation to transform validation data to training data?
>
> We thank the reviewer for the thoughtful review. Here are our responses.
>
> If the train and validation are identical, we would get a cosine similarity equal to 1, which is theoretical upper limits of data augmentation. However, the training and validation sets are non-overlapping and we are searching in a limited set of transformtions (e.g., rotate, shift, brightness, etc.) which is not capable of transforming training image to an images identical to the validation image. What we can achieve through data augmentation search is that we obtain a distribution of transformations that replicates the distribution of the images in the validation set. For example, we may obtain a distribution over various magnitudes of brightness transfomation that reproduces the distribution of lighting conditions in the validation dataset.
>
> > Could the author[s] provide some intuition on why using gradient matching for the search.
>
> As stated in Eqn. (10) in our submission, our work explicitly optimizes the gradient similarities with the average reward minus the standard deviation, which acts as a regularization term in our optimization formulation. In other words, our approach aims to maximize the gradient similarity while constraining its variance from growing too much. Therefore, our approach actually focuses **on two metrics**: the **mean value** and the **standard deviation** of the **improvement of gradient similarity**. As we will show below, optimizing the mean value of gradient similarity along the direction with low variance has achieved better performance.
>
> To clearly illustrate the relationship of these **two metrics** and the **final performance** across different numbers of layers as requested by the reviewer, we draw 3 plots: (1) the relationship between the number of layers of DeepAA and the **mean value** of gradient similarity improvement over the raw data (to remove the baseline gradient similarity of the raw data); (2) the relationship between the number of layers of DeepAA and the **standard deviation** of gradient similarity improvement over the raw data (to remove the baseline gradient similarity of the raw data); and (3) the relationship between the number of layers of DeepAA and the **final performance**. The reported accuracy is averaged over 4 independent runs. The mean and standard deviation of the gradient similarity improvement are calculated over 256 independently sampled raw images.
>
> Inspired by **reviewer BGUU**, we design a true augmentation baseline named “**DeepTA**” . In DeepTA, we stack multiple layers of TA (TrivialAugment) on the same augmentation space of DeepAA without default augmentations. In doing so, we can fairly compare the behavior of DeepAA and the baseline on the requested relationship across different numbers of layers.
>
> Given that we cannot insert plots in the comment, our plots can be accessed via [the anonymous link](https://figshare.com/s/3bf5f91de987d5ce82c2). From the figures we have the following observations:
>
> 1. For DeepTA, we observe from Figure (a) that the cosine similarity reaches the peak at the 5th layer, then stacking more layers will decrease the cosine similarity. From Figure (b) we see that the standard deviation increases considerably when stacking more layers.
> 2. For DeepAA, we observe that the cosine similarity increases consistently until it converges to identity transformation (see Figure 3(a) in the paper) at layer 6. As expected, the standard deviation does not grow as fast as DeepTA.
> 3. From Figure (c), both DeepAA and DeepTA reach peak performance with 6 layers of augmentation. We find that DeepAA achieves higher cosine similarity while maintaining lower standard deviation when converged at the 6th layer, which indicates that **maximizing the cosine similarity along the path with low variance** can lead to higher performance.
>
> >  If it will cause overfitting issues on training data?
>
> As stated above,  our approach explicitly optimizes the gradient similarities with the average reward minus the standard deviation, which acts as a regularization term in our optimization formulation.  The regularization term ensures that gradient similarity grows along the path with low variance, thus preventing augmentation policy from overfitting.

---

> ### Comment · Reviewer_9qhP · 2021-11-24
> **Response to Authors**
>
> Thanks to the authors for the detailed response. The response has addressed my concerns. I would highly recommend the authors add the informative augmentation policy comparison figure into the final version of the paper.

---

> > ### Author Response · Authors · 2021-11-26
> > **Thank you**
> >
> > We thank the reviewer for taking time to read our response. We are pleased to hear that our response addressed your concerns. We will include the augmentation policy comparison figure in the final version.

---

### Official Review · Reviewer_VFdc · 2021-11-02

**Correctness:** 3
**Technical Novelty And Significance:** 3
**Empirical Novelty And Significance:** 3
**Recommendation:** 8
**Confidence:** 3

**Main Review:**

## Strength
- The proposed method reduces the searching time for multi-layered augmentation policy by a large margin while achieving good performance.
- It is interesting to see that the latter augmentation layers converge to identify mapping (Fig3), which is a good indicator for the optimal number of augmentation layers.


## Weakness
- It is not well justified in the experiment section why formulating data augmentation search as a gradient matching problem is favorable. Even though the authors provide the similarity scores after stacking more augmentation layers in Tab6, the comparison of the cosine similarity of gradient between the proposed method and other augmentation methods is missing. Furthermore, the reason that higher cosine similarity of gradients leads to better final performance is missing. There exist a correlation between these two but the reviewer expect more analysis and insights for the causality.
- It is unclear whether the proposed method generalized to other tasks (eg. object detection)

**Summary Of The Paper:**

This paper proposed a learned data augmentation method. The proposed method is able to search for the augmentation policy with multiple layers by searching the next transformation operation conditioned on all previous augmentations. The authors also proposed to use cosine similarity between the gradients of the original and augmented data as guidance rather than e.g. the more commonly used classification validation loss.

**Summary Of The Review:**

One major weakness of this work is that the main idea of using cosine similarity of gradients as guidance for policy search is not well justified in the experiment section. Except for that, based on the experimental results that (1) the proposed method reduces the policy search time by a large margin while maintaining good performance compared to other SoTA augmentation methods, and that (2) the proposed method is applicable to multi-layer augmentation, overall the reviewer considers it a good paper.

---

> ### Author Response · Authors · 2021-11-20
> **Response to reviewer VFdc (1/1)**
>
> > It is not well justified in the experiment section why formulating data augmentation search as a gradient matching problem is favorable.
>
> We thank the reviewer for the thoughtful review. Here are our responses.
>
> As stated in Eqn. (10) in our submission, our work explicitly optimizes the gradient similarities with the average reward minus the standard deviation, which acts as a regularization term in our optimization formulation. In other words, our approach aims to maximize the gradient similarity while constraining its variance from growing too much. Therefore, our approach actually focuses **on two metrics**: the **mean value** and the **standard deviation** of the **improvement of gradient similarity**. As we will show below, optimizing the mean value of gradient similarity along the direction with low variance has achieved better performance.
>
> To clearly illustrate the relationship of these **two metrics** and the **final performance** across different numbers of layers as requested by the reviewer, we draw 3 plots: (1) the relationship between the number of layers of DeepAA and the **mean value** of gradient similarity improvement over the raw data (to remove the baseline gradient similarity of the raw data); (2) the relationship between the number of layers of DeepAA and the **standard deviation** of gradient similarity improvement over the raw data (to remove the baseline gradient similarity of the raw data); and (3) the relationship between the number of layers of DeepAA and the **final performance**. The reported accuracy is averaged over 4 independent runs. The mean and standard deviation of the gradient similarity improvement are calculated over 256 independently sampled raw images.
>
> Inspired by **reviewer BGUU**, we design a true augmentation baseline named “**DeepTA**” . In DeepTA, we stack multiple layers of TA (TrivialAugment) on the same augmentation space of DeepAA without default augmentations. In doing so, we can fairly compare the behavior of DeepAA and the baseline on the requested relationship across different numbers of layers.
>
> Given that we cannot insert plots in the comment, our plots can be accessed via [the anonymous link](https://figshare.com/s/3bf5f91de987d5ce82c2). From the figures we have the following observations:
>
> 1. For DeepTA, we observe from Figure (a) that the cosine similarity reaches the peak at the 5th layer, then stacking more layers will decrease the cosine similarity. From Figure (b) we see that the standard deviation increases considerably when stacking more layers.
> 2. For DeepAA, we observe that the cosine similarity increases consistently until it converges to identity transformation (see Figure 3(a) in the paper) at layer 6. As expected, the standard deviation does not grow as fast as DeepTA.
> 3. From Figure (c), both DeepAA and DeepTA reach peak performance with 6 layers of augmentation. We find that DeepAA achieves higher cosine similarity while maintaining lower standard deviation when converged at the 6th layer, which indicates that **maximizing the cosine similarity along the path with low variance** can lead to higher performance.

---

> ### Comment · Reviewer_VFdc · 2021-11-29
> **Response to Authors**
>
> The reviewer appreciates the additional experiments from the authors that well motivate the use of mean and standard deviation of gradient similarity. Therefore, the reviewer decides to keep the original rating.

---

### Official Review · Reviewer_3j37 · 2021-11-07

**Correctness:** 3
**Technical Novelty And Significance:** 3
**Empirical Novelty And Significance:** 3
**Recommendation:** 6
**Confidence:** 5

**Main Review:**

**Pros:**

- **P1.** **[more automated]** This work takes a further step towards automated machine learning by absorbing some default image transformations (flipping and cropping) into the operation set. The idea is natural, reasonable, and worth exploring.

- **P2.** **[relatively novel; enlightening]** The formulation of gradient matching is radically different from previous bilevel-optimization based works. Though it is not a new idea to perform gradient matching in meta-learning, this opens the avenue for automated data augmentation search, and could potentially be useful for other scenarios like self-supervised learning.


**Cons:**

- **C1.** **[missing details on other default operations]** Currently there are only two default operations (flipping and cropping) being searched. But there are also some other defaults like pixel value scaling (normalization) and color jittering. Are they still used by default in DeepAA? Or are there maybe some existing experimental results that include these two (norm and colorjitter) default augmentations?

- **C2.** **[correctness of statements]** A few statements are not well-supported, or require small changes to be made correct: The authors claim that DeepAA outperforms existing automated data augmentation methods, but it seems that some recent work is missing [1, 2]. The authors should add the comparisons to the paper for correctness. Though this might make the performances of DeepAA look weaker, it wouldn’t devalue the novelty and contributions.

- **C3.** **[reliability of new metric]** This work uses a new metric rather than the validation accuracy for augmentation search. However, the reliability of this metric is not well assessed. A straightforward way is to analyze the *correlation* between gradient similarities and final performances, *e.g.*, to evaluate different augmentation policies and to see whether the policies with higher gradient similarities would have better performances.

- **C4.** **[unclear experimental details]** I got confused with some details and have some questions regarding them:

  - **C4.1.** In Sec. 4, the paper introduces that sub-datasets are used to train the networks. Does this mean that the CNN networks are pretrained on these sub-datasets before conducting the augmentation search (like [3])? If this is true, why not to use the full set since the pretraining seems not so expensive?

  - **C4.2.** In Sec. 4.1, the paper indicates the policy is updated for 512 iterations. Does this mean that only 512 images are used to search augmentation policies? Since Eq. (12) shows only one single data item is used for each policy update. If it's true, why not to use more training samples to further boost the policy? It would be insightful if a discussion is included on the choice of this hyper-parameter.

**Minor Problems:**

- [Sec. 1.] "an effective *metrics*" -- "*metric*"
- [Sec. 3.1.] "where the augmentation policies *is* shallow" -- "*are* shallow".
- [Sec. 3.3.3.] "transformations that *performs* good" -- "*perform*"
- Figure 2. and 3. are not vectorized and look blurry. Using vector graphics (like Figure 1.) could be better.

---------------------------

[1] Zhang, Xinyu, et al. "Adversarial AutoAugment." International Conference on Learning Representations. 2019.

[2] Tian, Keyu, et al. "Improving Auto-Augment via Augmentation-Wise Weight Sharing." Advances in Neural Information Processing Systems 33 (2020).

[3] Lim, Sungbin, et al. "Fast autoaugment." Advances in Neural Information Processing Systems 32 (2019): 6665-6675.






**Summary Of The Paper:**

A more "automated" augmentation search method is proposed. The main contributions are:

- Default transformations like flipping and cropping are regarded as new augmentation candidates which could be optimized, making the algorithm more automated.
- The gradient matching formulation is borrowed and seems to be effective for augmentation search.
- MC sampling and a progressive search pipeline are used to reduce the computational complexity.

**Summary Of The Review:**

I tend to give a rejection for now. This work is well motivated and has a natural goal. The formulation of gradient matching brings something new to this area. However, there are some flaws like claims with minor issues (**C2**) and missing details (**C1, C4**). Some claims needs to be updated and some details need further explanations. Furthermore, I have some questions about the reliability (**C3**) which may affect my recommendation as well. If all of these issues are solved, I'll raise my score.

--------
*After rebuttal: my main concerns are well addressed, hence I recommend to accept this work. Besides, the manuscript can be improved by providing more discussion and analysis. See the response to authors.

---

> ### Author Response · Authors · 2021-11-19
> **Response to reviewer 3j37 (1/4)**
>
> We thank the reviewer for the thoughtful review. Here are our responses.
>
> > (1/4) Missing details on other default operations
>
> We applied pixel value normalization (i.e., normalize RGB channels with mean and standard deviation) on every image. However, such normalization should not be considered as an augmentation operation. This is because it is applied deterministically to all the data, which does not increase data diversity. We do not include color jittering which is used as a default augmentation in previous works. This is because color jittering is similar to the augmentation operations such as color, brightness, and contrast, which are already included in our DeepAA augmentation space.

---

> ### Author Response · Authors · 2021-11-19
> **Response to reviewer 3j37 (2/4)**
>
> > (2/4) correctness of statements
>
> We apologize for missing the two references pointed out by the reviewer (i.e., AdvAA and AWS). To correct this mistake, we need to include the results of AdvAA and AWS into our paper and have a direct comparison to DeepAA.
>
> [1] Zhang, Xinyu, et al. "Adversarial AutoAugment." International Conference on Learning Representations. 2019.
>
> [2] Tian, Keyu, et al. "Improving Auto-Augment via Augmentation-Wise Weight Sharing." Advances in Neural Information Processing Systems 33 (2020).

---

> ### Author Response · Authors · 2021-11-20
> **Response to reviewer 3j37 (3/4)**
>
> > (3/4) Reliability of new metric
>
> As stated in Eqn. (10) in our submission, our work explicitly optimizes the gradient similarities with the average reward minus the standard deviation, which acts as a regularization term in our optimization formulation. In other words, our approach aims to maximize the gradient similarity while constraining its variance from growing too much. Therefore, our approach actually focuses **on two metrics**: the **mean value** and the **standard deviation** of the **improvement of gradient similarity**. As we will show below, optimizing the mean value of gradient similarity along the direction with low variance has achieved better performance.
>
> To clearly illustrate the relationship of these **two metrics** and the **final performance** across different numbers of layers as requested by the reviewer, we draw 3 plots: (1) the relationship between the number of layers of DeepAA and the **mean value** of gradient similarity improvement over the raw data (to remove the baseline gradient similarity of the raw data); (2) the relationship between the number of layers of DeepAA and the **standard deviation** of gradient similarity improvement over the raw data (to remove the baseline gradient similarity of the raw data); and (3) the relationship between the number of layers of DeepAA and the **final performance**. The reported accuracy is averaged over 4 independent runs. The mean and standard deviation of the gradient similarity improvement are calculated over 256 independently sampled raw images.
>
> Inspired by **reviewer BGUU**, we design a true augmentation baseline named “**DeepTA**” . In DeepTA, we stack multiple layers of TA (TrivialAugment) on the same augmentation space of DeepAA without default augmentations. In doing so, we can fairly compare the behavior of DeepAA and the baseline on the requested relationship across different numbers of layers.
>
> Given that we cannot insert plots in the comment, our plots can be accessed via [the anonymous link](https://figshare.com/s/3bf5f91de987d5ce82c2). From the figures we have the following observations:
>
> 1. For DeepTA, we observe from Figure (a) that the cosine similarity reaches the peak at the 5th layer, then stacking more layers will decrease the cosine similarity. From Figure (b) we see that the standard deviation increases considerably when stacking more layers.
> 2. For DeepAA, we observe that the cosine similarity increases consistently until it converges to identity transformation (see Figure 3(a) in the paper) at layer 6. As expected, the standard deviation does not grow as fast as DeepTA.
> 3. From Figure (c), both DeepAA and DeepTA reach peak performance with 6 layers of augmentation. We find that DeepAA achieves higher cosine similarity while maintaining lower standard deviation when converged at the 6th layer, which indicates that **maximizing the cosine similarity along the path with low variance** can lead to higher performance.

---

> ### Author Response · Authors · 2021-11-20
> **Response to reviewer 3j37 (4/4)**
>
> > (4/4)  Unclear experimental details. <1> Does this mean that the CNN networks are pretrained on these sub-datasets before conducting the augmentation search? <2> Why not to use the full set since the pretraining seems not so expensive? <3> The paper indicates the policy is updated for 512 iterations. Does this mean that only 512 images are used to search augmentation policies? Since Eq. (12) shows only one single data item is used for each policy update. If it's true, why not to use more training samples to further boost the policy? It would be insightful if a discussion is included on the choice of this hyper-parameter.
>
> 1. Yes. We followed the settings in Fast AutoAugment (Fast AA) [1], where the augmentation policy is searched on a proxy model pretrained on a subset of the full dataset.
>
> 2. We can pretrain the model using the full set, but we need to terminate pretrain training at an early stage. More specifically, according to [2], data augmentation affects early learning dynamics but matters little near convergence. Thus we need to keep the pretrained model in the early training phase instead of near convergence. One way of keeping the model in the early training phase is to train the model with a small subset of the data until it converges on such a subset. Another way of doing so is to train the model on the full dataset but terminate training at an early stage. We found that both ways lead to similar results as long as the model is at the early stage of training.
>
> 3.
>     * We do use more than 512 images. Specifically, we optimize the policy of each layer for 512 iterations. The reward \tilde{r}^k(x) in Eqn. (9) is calculated using a single image augmented multiple times, while for each iteration we update the policy following Eqn. (12) using the reward r^k in  Eqn. (10), which is estimated by Monte Carlo averaging over 16 different images. Therefore, we use 512x16=8192 images for augmentation policy search for each layer.
>      * We find that the trajectory of cosine similarity converges within 512 iterations for each layer, so extending search iterations over 512 iterations is not necessary.
>
>
> [1] Lim, Sungbin, et al. "Fast autoaugment." Advances in Neural Information Processing Systems 32 (2019): 6665-6675.
>
> [2] Time matters in regularizing deep networks: Weight decay and data augmentation affect early learning dynamics, matter little near convergence. Advances in Neural Information Processing Systems, 32:10678–10688, 2019.

---

> ### Comment · Reviewer_3j37 · 2021-11-23
> **Response to Authors**
>
> The supplementary results and clarifications are compelling. My concerns are well addressed. No further questions. I've increased the score and, lastly, I would suggest the authors add the details in discussion into their final paper, especially the reliability assessment (Response 3/4).

---

> > ### Author Response · Authors · 2021-11-26
> > **Thank you**
> >
> > We thank the reviewer for taking time to read our response. We are glad that our response addressed your concerns. We will add the discussion, including the reliability assessment, in the final version.

---

### Public Comment · ~Long_M._Luu1 · 2022-02-22
**A great paper. Looking for the code release, and also a small typo**

Dear authors,

Thanks for your paper. Your work is very interesting. AutoAugment code was never released to the best of my knowledge, thus your paper contributes a lot to the community, and I am looking forward to the code release.

There is one typo I found: in Figure 4(c), the Y-axis label is wrong: it should be "Mean accuracy over the course of optimization", but instead now it has the label of Figure 4(a).

Aside from that, congratulations!

---

> ### Public Comment · ~Mi_Zhang1 · 2022-02-22
> **Code release**
>
> Dear Long,
>
> Thank you for your interest to our work and pinpointing the typo.
>
> Our code is released at our github repo at: https://github.com/MSU-MLSys-Lab/DeepAA
>
> Hope this is helpful to your own research!

---

### Decision · Program_Chairs · 2022-01-20

**Decision:**

Accept (Poster)

**Comment:**

We appreciate the authors for engaging in discussions with the reviewers and providing further experimental results to clarify and address the concerns raised by them in their original reviews, leading to changes in some of the recommendations.

While the (revised) paper with the clarifications and new results incorporated are more ready for publication, some outstanding concerns should preferably be addressed to further enhance its quality.

The authors are highly recommended to take into consideration all the comments and suggestions of the reviewers to further revise their paper to make it a scholarly work to contribute to the ICLR and the more general ML community.